# Synergistic Effect between *Trichoderma virens* and *Bacillus velezensis* on the Control of Tomato Bacterial Wilt Disease

**You Zhou, Laying Yang, Jun Wang, Lijia Guo and Junsheng Huang \*** 

Environment and Plant Protection Institute, Chinese Academy of Tropical Agricultural Sciences, No. 4, West Xueyuan Road, Haikou 571101, China; youzhou168@catas.cn (Y.Z.); layingyang@catas.cn (L.Y.); wangjun@catas.cn (J.W.); guolijia@catas.cn (L.G.)
\* Correspondence: huangjunsheng@catas.cn

**Abstract:** (1) Background: *Ralstonia solanacearum* causes tomato bacterial wilt disease, one of the most serious tomato diseases. As the combination of *Trichoderma virens* (Tvien6) and *Bacillus velezensis* (X5) was more effective at controlling tomato bacterial wilt disease than a single agent, we investigated the synergistic effect of Tvien6 and X5 in controlling this disease; (2) Methods: The disease incidence, plant heights and weights, relative chlorophyll content (SPAD values), defensive enzymes (PPO, POD, and SOD) activities, and metabolome were estimated among four treatment groups (BR treatment, X5 + *R. solanacearum* (RS-15); TR treatment, Tvien6+ RS-15; TBR treatment, Tvien6 + X5 + RS-15; and R treatment, RS-15); (3) Results: The R treatment group had the highest disease incidence and lowest plant heights, plant weights, SPAD values, defensive enzyme activities, and D-fructose and D-glucose contents; the TBR treatment group had the lowest disease incidence and highest plant heights, plant weights, SPAD values, defensive enzyme activities, and D-fructose and D-glucose contents; (4) Conclusions: The results revealed that Tvien6 and X5 can both individually promote tomato plant growth, increase leaf chlorophyll content, enhance defensive enzyme activities, and induce the accumulation of D-fructose and D-glucose; however, they were more effective when combined.

**Keywords:** tomato bacterial wilt disease; *Trichoderma virens*; *Bacillus velezensis*; synergistic effect



## 1. Introduction

Tomato bacterial wilt disease, caused by *Ralstonia solanacearum*, is a soil-borne disease that severely endangers the tomato industry [1,2]. Bacterial wilt disease is a systemic disease that causes tomato root necrosis and vascular bundle browning, which can inhibit tomato nutrient absorption and cause plant death. At present, the control of tomato bacterial wilt in the field is mainly based on chemical pesticides, but the effects are sometimes insignificant and can easily cause soil pollution.

Biological control of plant diseases is an ecologically and environmentally friendly method that can guarantee the sustainable development of agriculture [3,4]. *Bacillus* spp. and *Trichoderma* spp. have been successfully used to control a variety of plant bacterial and fungal diseases, and their biocontrol mechanisms include producing antagonistic substances, producing parasitic/reparasitic substances, competing with pathogens for nutrition and space, and inducing plant resistance, among others [5–9].

Many studies have shown that the combination of two or more biocontrol agents to control diseases is more effective than a single agent [10–15]. One explanation for the synergistic effects of multiple biocontrol agents may be in the mixing of agents with different mechanisms of action (such as competition and the production of antagonistic or parasitic/reparasitic substances) [16]. Guetsky et al. (2002) considered that the combined use of *Pichia guilermondii* and *Bacillus mycoides* showed a synergistic antagonist effect on *Botrytis cinerea* as *P. guilermondii* could compete with *B. cinerea* for nutrients, and

*Bacillus mycoides* could produce antagonistic substances to destroy *B. cinerea* cells [17]. Xu and Jeger (2013) demonstrated that the control effect of the combined use of a mycoparasitic biocontrol agent with a competitive agent on pathogens is better than that of two mycoparasitic or two competitive biocontrol agents [18].

It is worth noting that the antagonistic substances produced by different types of biocontrol agents are not the same, and mixing their products can improve the antagonism of pathogens. The study by Jimtha John et al. (2016) showed that the antagonistic effect of a mixed aseptic fermentation broth of *Bacillus* sp. and *Trichoderma* sp. on *Pythium myriotylum* was better than that of *Bacillus* sp. or *Trichoderma* sp. aseptic fermentation broth alone [19]. Woo et al. (2002) found that the antagonistic ability of a mixture of cell wall degrading enzymes of *Trichoderma virens* with lipodepsipeptides produced by *Pseudomonas syringae* was higher than that of using only cell wall degrading enzyme or lipodepsipeptides [20].

Some studies have shown that the combination of *Bacillus* spp. and *Trichoderma* spp. to control grapevine gray mold (caused by *Botrytis cinerea*), ginger soft rot (caused by *Pythium myriotylum*), tobacco bacterial wilt (caused by *Ralstonia solanacearum*), tobacco damping-off (caused by *Pythium aphanidermatum*), tobacco frogeye leaf spot (caused by *Cercospora nicotiana*), and cucumber damping-off (caused by *Rhizoctonia solani*) diseases is more effective than using only *Bacillus* sp. or *Trichoderma* sp. [14,19,21,22].

It has been found that a range of phenomena are associated with plants treated with biocontrol agents, some of which are involved in promoting growth, improving chlorophyll, and enhancing disease resistance of plants [8,23–25].

Various mechanisms have been used to explain plant growth promotion associated with *Bacillus* spp. and *Trichoderma* spp., including enhancing nutrient uptake, influencing the contents of phytohormones, and increasing the rate of carbohydrate metabolism and photosynthesis [24,26–28]. Biocontrol agents increase plant growth and have also been reported for many vegetables, such as tomato, cucumber, and lettuce [29–31].

Some previous studies have clarified the importance of POD, PPO, and SOD in plant resistance to diseases [32,33]. Changes in peroxidase (POD), polyphenol oxidase (PPO), and superoxide dismutase (SOD) activities are important physiological indicators reflecting plant disease resistance, and biocontrol agents can induce their activities, which enable plants to acquire induced systemic resistance (ISR) and make the plant resistant to phytopathogens [6,34–37].

Leaf chlorophyll concentration is an important indicator of photosynthetic capacity and plant health, and chlorophyll loss has been reported to be associated with environmental stress on plants [38,39]. The SPAD value is proportional to the chlorophyll content in the sample [40–42]. Most studies have shown that *Bacillus* and *Trichoderma* could significantly increase the leaf SPAD value of barley, wheat, lettuce, tomato, zucchini, and pepper, and the increase in SPAD value was associated with growth promotion [43–46].

Changes in metabolites can be regarded as the organism's response to changes in the external environment [47,48]. Murti et al. (2021) showed that leucine was significantly upregulated and valine was significantly downregulated when tomato was infected with *Ralstonia solanacearum*, and they are regarded as important metabolites in the defense mechanisms of tomato against bacterial wilt [49]. Zeiss et al. (2019) showed that after bacterial wilt susceptible and resistant tomato cultivars were infected with *Ralstonia solanacearum*, the contents of some metabolites of the phenylpropanoid pathway were different, which indicated that the phenylpropanoid pathway is of prime importance in tomato defense after infection with *Ralstonia solanacearum*, and that flavonoids and hydroxycinnamic acids are representative metabolites [50]. In addition, many studies have also shown that under the stimulation of exogenous microorganisms, plants can induce disease resistance by changing their metabolite contents [5,8,51,52].

The present investigations were carried out to determine the influence of the combined use of *Trichoderma virens* (Tvien6) and *Bacillus velezensis* (X5) on inhibiting bacterial wilt disease, and on plant growth and the physiological parameters of tomato. Therefore, we estimated the disease incidence, plant height, plant weight, defensive enzymes (PPO, POD,

and SOD) activities, SPAD value, and metabolite contents of tomatoes challenged with *Ralstonia solanacearum* and treated with Tvien6 and X5 individually and in combination, which could help to understand the synergistic effect of Tvien6 and X5 in controlling tomato bacterial wilt disease.

## 2. Materials and Methods

### 2.1. Biocontrol Agents, Pathogen, and Cells/Conidia Suspension Collection

The *Trichoderma virens* (Tvien6) and *Bacillus velezensis* (X5) strains utilized in this study were obtained from the Environment and Plant Protection Institute at the Chinese Academy of Tropical Agricultural Sciences. The pathogen *Ralstonia solanacearum* (RS-15) was isolated from naturally infected tomato plants with bacterial wilt symptoms. *Ralstonia solanacearum* was identified by morphological characteristics and three pairs of specific primers (fliC-F/fliC-R, pehA#3/pehA#6, and 759/760) [53–55].

Tvien6 were grown in the dark for 7 days on Petri dishes containing PDA medium at 26 °C. The surfaces of the hypha were scraped with a sterile coating rod to remove the conidia, the hypha were filtered out with sterile four-layer gauze, and the resulting suspension was diluted with sterile distilled water to a concentration of $1 \times 10^6$ mL$^{-1}$.

X5 was grown in 150 mL of LB liquid medium in flasks and shaken for 36 h at 150 rpm at 37 °C in the dark. RS-15 was grown in 150 mL of nutrient sucrose broth in flasks [56] and shaken for 48 h at 150 rpm at 30 °C in the dark. The X5 and RS-15 cells were then collected by centrifugation (X5: 4500 rpm, 15 min; RS-15: 7500 rpm, 8 min), washed with sterile distilled water, centrifuged again (X5: 4500 rpm, 15 min; RS-15: 7500 rpm, 8 min), and diluted with sterile distilled water to a concentration of $1 \times 10^6$ mL$^{-1}$.

### 2.2. Effect of Trichoderma virens and Bacillus velezensis Culture Filtrates on Ralstonia solanacearum Growth

A 150 μL suspension of RS-15 cells ($1 \times 10^6$ mL$^{-1}$) was spread evenly on a TTC plate [4]. Then, holes were punched in the plate with a sterile puncher, and 200 μL of culture filtrates from Tvien6 and X5 cultured alone or cocultured together were added to the holes. The culture filtrates were obtained as follows: for the co-culture, 2 mL each of X5 and Tvien6 cells/conidia suspensions were added to 200 mL of LBPD liquid medium (containing 100 mL liquid LB and 100 mL liquid PD medium); for the individual cultures, 4 mL of X5 or Tvien6 cells/conidia suspensions were added to 200 mL of LBPD liquid medium. Three replicates of each culture were carried in the dark out at 28 °C with shaking at 150 rpm for 96 h. The culture filtrates were centrifuged at 3500 rpm for 10 min and passed through a filter membrane (pore size 0.22 μm).

### 2.3. Pot Experiments

Soil (red soil: cow dung: sand in a 3:0.5:0.5 ratio (W:W:W)) was autoclaved at 105 °C for 2 h over three consecutive days. The tomato seedlings (*Lycopersicon esculentum*, cv. Qianxi) were planted in pots/basins in a greenhouse at 30 °C/day, 27 °C/night, and 80% RH, and they were watered with a 1% strength compound fertilizer solution (N:P:K = 15:15:15) every 5 days. Tomato seedlings 8–10 cm tall were selected for experiments. Four experimental treatments were carried out as follows: BR treatment group plants were treated with X5 and RS-15; TR treatment group plants were treated with Tvien6 and RS-15; TBR treatment group plants were treated with Tvien6, X5, and RS-15; R treatment group plants (as control) were treated with the RS-15 pathogen only.

The BR, TR, and TBR treatment groups were drenched with 350 mL of Tvien6 and X5 cells/conidia suspensions ($1 \times 10^6$ mL$^{-1}$) individually or in combination. After 3 days, 350 mL of the RS-15 pathogenic cells suspension ($1 \times 10^6$ mL$^{-1}$) was added to the BR, TR, TBR, and R treatment groups.

Pots/basins were arranged in a randomized block design. For disease incidence, plant height, plant weight, and relative chlorophyll content (SPAD values) measurements, each treatment had three replicates, with 15 pots per replicate, and each pot contained 2

seedlings. For enzyme assays and GC-MS analysis, each treatment had three replicates, with 5 basins per replicate, and each basin contained 6 seedlings.

### 2.3.1. Disease Incidence, Relative Chlorophyll Content, Plant Height, and Plant Weight

Disease incidence, plant height, plant weight, and SPAD values were measured after 15 days of BR, TR, TBR, and R (as control) treatment groups with RS-15. Three replicates were conducted for each treatment, and each replicate contained 30 seedlings.

Disease incidence was calculated according to the following equation:

$$\text{Disease incidence} = \left( \frac{\text{Number of infected plants}}{\text{Total number of plants}} \right) * 100$$

Twelve tomato plants were randomly selected for each treatment, and the height of the plant (from the base of the stem to the growth point) and plant weight were measured. The tomato plants that were measured for height were then washed, the surface water was removed, and the weight of the whole plant was measured. Twelve replicates were conducted for each treatment, and each replicate contained 1 tomato plant.

For each treatment, 5 leaves of the same age from 5 different tomato plants were selected to measure the relative chlorophyll content (SPAD values) using a chlorophyll meter (SPAD-502Plus, Konica Minolta, Japan). A tomato leaf was divided into four sections, and every leaf measurement was an average of the four leaf sections' SPAD-502Plus readings. Each treatment had five replicates, and per replicate contained 1 leaf.

Statistical data analysis of disease incidence, plant height, plant weight, and SPAD values was performed using analysis of variance (ANOVA) and least significant difference (LSD) with open-source software tools (https://www.omicshare.com/tools/Home/Soft/letter_sig, accessed on 8 August 2021).

### 2.3.2. Enzyme Assays

The enzymatic assays were performed 0, 3, 6, and 9 d after pathogen inoculation for BR, TR, TBR, and R (as a control) treatment groups.

For measuring superoxide dismutase (SOD), peroxidase (POD), and polyphenol oxidase (PPO) activities, 0.3 g of tomato stem tissue was homogenized with 5 mL of ice-cold 50 mM phosphate buffer (pH 7.0) containing 2% polyvinylpyrrolidone (*w/v*) using a prechilled mortar and pestle. All steps of the extract were carried out at 2–4 °C. The homogenates were centrifuged at 12,000× *g* at 4 °C for 20 min, and the obtained supernatant was used for the assays of enzymatic activity.

SOD activity was assayed by measuring the ability to inhibit the photochemical reduction in p-nitro blue tetrazolium (NBT) at 560 nm using the method of Singh et al. (2013) [57]. Catechol was used as a substrate for determining PPO activity at 420 nm [58]. POD activity was evaluated at 470 nm as described by Song et al. (2011) [59] with guaiacol as the substrate.

Three replicates were conducted for each treatment, and each replicate contained 1 leaf of the same age. The statistical data analysis of SOD, POD, and PPO activities was performed using analysis of variance (ANOVA) and least significant difference (LSD) with open-source software tools (https://www.omicshare.com/tools/Home/Soft/letter_sig, accessed on 8 August 2021).

### 2.3.3. GC-MS Analysis of Different Treatments

Extraction and Derivatization

Tomato stems metabolites of BR, TR, TBR, and R (as control) treatment groups were extracted and derivatized according to the methods described by Lisec et al. (2006) [60] and Wang et al. (2015) [61] with some modifications. Six replicates were conducted for each treatment, and each replicate contained 1 tomato stem. All samples in the study were in random order. Tomato stems were quickly ground by liquid nitrogen, and 100 mg of the tomato stem powder was quickly weighed and placed in a 2 mL centrifuge tube. Then,

1.4 mL methanol (100%) (precooled at −20 °C) and 60 µL of ribitol (0.2 mg·mL$^{-1}$) were added to the tube as an internal quantitative standard. The sample was then vortexed for 60 s, treated with ultrasonication for 30 min at 70 °C, incubated at 70 °C for 15 min at 950 rpm, and centrifuged for 10 min at 11,000× *g*, after which 150 µL of the supernatant was transferred to a new 1.5 mL tube and vacuum-dried in a vacuum freeze dryer. Methanol was used as a blank sample.

A total of 40 µL of methoxyamine hydrochloride dissolved in pyridine (20 mg·mL$^{-1}$) was added to the sample tube, followed by shaking at 250 rpm for 2 h at 37 °C. Next, the sample was centrifuged for 30 s at 2200× *g*, 70 µL N,O-bis (trimethylsilyl) trifluoroacetamide (containing 1% trimethylchlorosilane) was added, and the tube was shaken again at 250 rpm for 60 min at 70 °C. Finally, the samples were cooled to room temperature, centrifuged for 30 s at 2200× *g*, and filtered through a 0.22 µm membrane before analysis.

GC-MS Analysis

A 7890A-5975C GC-MS system (Agilent Technologies, Santa Clara, CA, USA) with an HP-5ms capillary column (30 m × 0.25 mm × 0.25 µm) (Agilent J & W Scientific, Folsom, CA, USA) was used to analyze the samples. All samples were continuously, automatically, and randomly injected into the GC column at a constant flow rate of 1 mL·min$^{-1}$ with high-purity helium. The oven temperature was maintained at 80 °C for 2 min, increased to 300 °C at a rate of 5 °C/min and kept at this temperature for 1 min. The injector and electron impact ion source temperature was set to 250 °C, and electron impact (70 eV) mass spectra were recorded with a scanning range of 45–600 *m/z*.

The *m/z* peaks representing the mass to charge ratio characteristics of the metabolites were compared with those in the NIST mass spectrum database (National Institute for Standards and Technology). Principal component analysis (PCA) and heat mapping were conducted using open-source software tools (https://www.omicstudio.cn/tool/13; https://www.omicshare.com/tools/Home/Soft/heatmap, accessed on 25 November 2020) based on the relative content of metabolites, and the relative content of metabolites was calculated according to the following equation [62]:

$$\text{Relative content of metabolite} = \text{Peak area of metabolite} \div \text{Peak area of ribitol}$$

Statistical analysis of metabolites was performed using *t*-tests with open-source software tools (https://www.omicshare.com/tools/Home/Soft/mul_groups_sig, accessed on 25 November 2020).

## 3. Results

*3.1. Effect of Trichoderma virens and Bacillus velezensis on Pathogen and Tomato Growth and Disease Incidence*

The results showed that the co-cultured culture filtrates of Tvien6 and X5 significantly reduced *Ralstonia solanacearum* growth relative to Tvien6 or X5 alone (Figure 1a).

The disease incidence of the R treatment group (as a control) was significantly higher than that of the BR, TR, and TBR groups. The disease incidences of the TR and BR treatment groups were 42.22 ± 4.16% and 35.56 ± 5.67%, respectively, while the two biocontrol agents in combination (TBR treatment groups) resulted in a disease incidence of only 18.89 ± 1.57%; disease incidence with the pathogen alone (R treatment group) was 68.89 ± 5.67% (Table 1). Compared with the R treatment group, the plant heights of the BR, TR, and TBR treatment groups significantly increased by 10.27%, 7.59%, and 25.02%, respectively (Table 1). Similarly, compared with the R treatment group, the plant weights of the BR, TR, and TBR treatment groups significantly increased by 17.99%, 10.83%, and 35.02%, respectively (Table 1). Finally, the tomato plant SPAD values (relative chlorophyll content) of the BR and TBR treatment groups significant increased by 11.09% and 21.67%, respectively, compared with the R treatment group (Table 1).

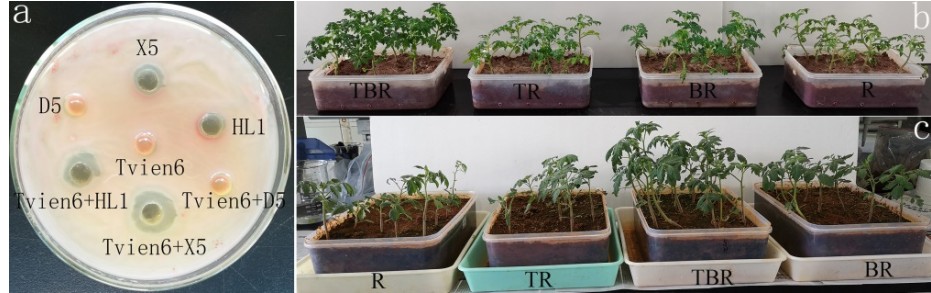

**Figure 1.** Effect of *Trichoderma virens* (Tvien6) and *Bacillus velezensis* (X5) on *Ralstonia solanacearum* (RS-15) and tomato plants. (**a**) In vitro antagonistic activity of single culture and cocultured filtrates of Tvien6 and X5 against *Ralstonia solanacearum*. D5 and HL1 are *Bacillus velezensis*, but they were not used in this study; (**b**,**c**) Effect of Tvien6 and X5 on tomato plant growth; the treatments were as follows: R = RS-15; TR = Tvien6 + RS-15; BR = X5 + RS-15; and TBR = Tvien6 + X5 + RS-15. For (**b**), three replicates were conducted for each treatment, with five basins per replicate, and each basin contained six seedlings. For (**c**), three replicates were conducted for each treatment, with five basins per replicate, and each basin contained 15 seedlings.

**Table 1.** Effect of *Trichoderma virens* and *Bacillus velezensis* on the plant height, plant weight, SPAD values, and disease incidence of tomatoes.

|  | Disease Incidence (%) | SPAD Value | Plant Height (cm) | Plant Weight (g) |
|---|---|---|---|---|
| BR | 35.56($\pm$6.94) b | 39.06($\pm$1.11) b | 25.87($\pm$0.95) b | 13.18($\pm$0.77) b |
| TR | 42.22($\pm$5.09) b | 37.24($\pm$1.33) bc | 25.24($\pm$0.79) b | 12.38($\pm$1.08) b |
| TBR | 18.89($\pm$1.92) c | 42.78($\pm$1.65) a | 29.33($\pm$0.89) a | 15.94($\pm$0.80) a |
| R | 68.89($\pm$6.94) a | 35.16($\pm$1.22) c | 23.46($\pm$1.06) c | 11.17($\pm$1.02) c |

Note: The data in the table are the mean $\pm$ SD value; different lowercase letters in the same column indicate that there were significant differences ($p < 0.05$) among different treatments. The treatments were as follows: R = *Ralstonia solanacearum* (RS-15); TR = *Trichoderma virens* (Tvien6) + RS-15; BR = *Bacillus velezensis* (X5) + RS-15; and TBR = Tvien6 + X5 + RS-15.

The results from the pot experiments indicated that Tvien6 and X5 increased the plant height, plant weight and SPAD values of the tomatoes and significantly decreased the disease incidence of tomato bacterial wilt disease, and the effect was more obvious with the combined use of the two strains (Table 1, Figure 1b,c).

*3.2. Enzyme Assays*

The PPO activity ranged from 31.34 $\pm$ 2.08 to 99.17 $\pm$ 2.64 U·min$^{-1}$·g$^{-1}$ FW among four treatment groups (Figure 2a). The PPO activities of the TBR and TR treatment groups showed an increasing trend from days 0 to 9, and the maximum PPO activities of the TBR and TR treatment groups were 99.17 $\pm$ 2.64 and 79.12 $\pm$ 0.56 U·min$^{-1}$·g$^{-1}$ FW, respectively. The PPO activity of the BR treatment group showed an increasing trend from days 0 to 6 and then showed a decreasing trend, and the maximum PPO activity was 84.43 $\pm$ 3.59 U·min$^{-1}$·g$^{-1}$ FW. In the R treatment groups, the peak PPO activity (76.95 $\pm$ 2.67 U·min$^{-1}$·g$^{-1}$ FW) was observed on day 3, which showed a decreasing trend then a slight increasing trend after day 6. The maximum PPO activity (99.17 $\pm$ 2.64 U·min$^{-1}$·g$^{-1}$ FW) among four treatment groups was recorded in the TBR treatment group on day 9. Compared with the R treatment group, PPO activity was significantly increased in the TR, BR, and TBR treatment groups on days 0, 6, and 9.

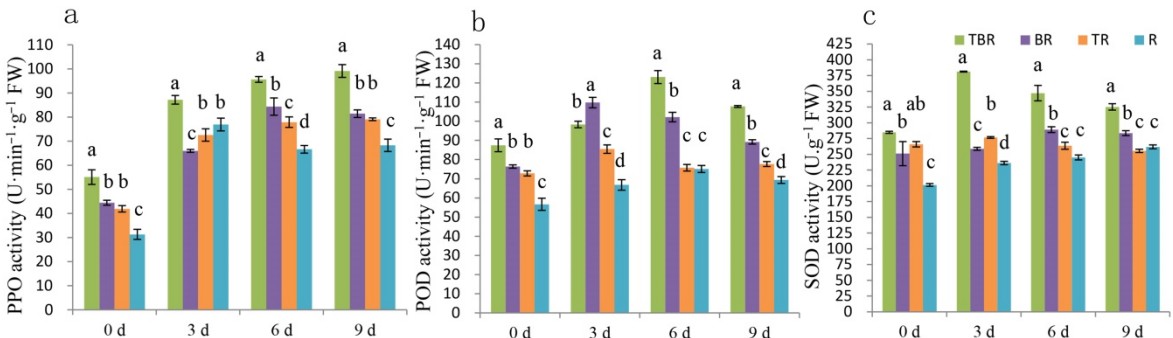

**Figure 2.** Effect of *Trichoderma virens* and *Bacillus velezensis* on induction PPO, POD, and SOD activities in tomato plants. The treatments were as follows: R = *Ralstonia solanacearum* (RS-15); TR = *Trichoderma virens* (Tvien6) + RS-15; BR = *Bacillus velezensis* (X5) + RS-15; and TBR = Tvien6 + X5 + RS-15. The values of SOD (**c**), POD (**b**), and PPO (**a**) activities are shown as the mean of three replications ± standard deviation (SD); different lowercase letters on the columns indicate that there were significant differences ($p < 0.05$) among different treatments.

The variation in POD activity ranged from $56.66 \pm 3.17$ to $123.11 \pm 3.39$ U·min$^{-1}$·g$^{-1}$ FW among these four treatment groups (Figure 2b). The POD activity of the TBR and R treatment groups showed an increasing trend from days 0 to 6, and showed a decreasing trend after day 6. The maximum POD activities of the TBR and R treatment groups were $123.11 \pm 3.39$ and $75.11 \pm 1.82$ U·min$^{-1}$·g$^{-1}$ FW, respectively. The POD activity of the BR and TR treatment groups showed an increasing trend from days 0 to 3 and showed a decreasing trend after day 3, but the POD activity of the TR treatment group showed a slight increasing trend after day 6. The maximum POD activities of the BR and TR treatment groups were $109.77 \pm 2.71$ and $85.49 \pm 2.17$ U·min$^{-1}$·g$^{-1}$ FW, respectively. The maximum POD activity ($123.11 \pm 3.39$ U·min$^{-1}$·g$^{-1}$ FW) among four treatment groups was recorded in the TBR treatment group on day 6. Compared with the R treatment group, POD activity was significantly increased in the TR, BR, and TBR treatment groups on days 0, 3, and 9.

The SOD activity varied from $201.3 \pm 2.13$ to $381.22 \pm 0.7$ U·g$^{-1}$ FW among four treatment groups (Figure 2c). The SOD activity of the TBR and TR treatment groups showed an increasing trend from days 0 to 3 and showed a decreasing trend after day 3. The maximum SOD activities of the TBR and TR treatment groups were $381.22 \pm 0.7$ and $276.75 \pm 1.32$ U·g$^{-1}$ FW, respectively. In the BR treatment group, the SOD activity showed an increasing trend from days 0 to 6 and showed a decreasing trend after day 6, and the peak SOD activity ($289.15 \pm 4.55$ U·g$^{-1}$ FW) was observed on day 6. The SOD activity in the R treatment group increased from days 0 to 9, and the maximum activity was $261.76 \pm 3.14$ U·g$^{-1}$ FW. The maximum SOD activity ($381.22 \pm 0.7$ U·g$^{-1}$ FW) among four treatment groups was recorded in the TBR treatment group on day 3. Compared with the R treatment group, SOD activity was significantly increased in the TR, BR, and TBR treatment groups on days 0 and 3.

In most cases, the order of PPO, POD, and SOD activity among four treatment groups was TBR treatment group > BR and TR treatment group > R treatment group. It can be seen from the results that both Tvien6 and X5 can increase PPO, POD, and SOD activities; however, they more strongly induce the activity of these enzymes when in combination, which shows that Tvien6 and X5 synergistically effect the induction of defense enzymes activity.

*3.3. GC-MS Analysis*

Only 16 metabolites were obtained from four treatments by GC-MS. Of these, 13 were identified unambiguously. In the principal component analysis (PCA) plot, there were significant separations between the metabolite profiles of the biocontrol agent treatment groups and the pathogen only treatment group. The four treatments could be discriminated by PC1 (46.23%) and PC2 (25.1%) (Figure 3a). However, the separation between the BR

and TR treatment groups was less distinct. The heatmap clearly showed that the relative metabolite contents were different among the four treatment groups (Figure 3b).

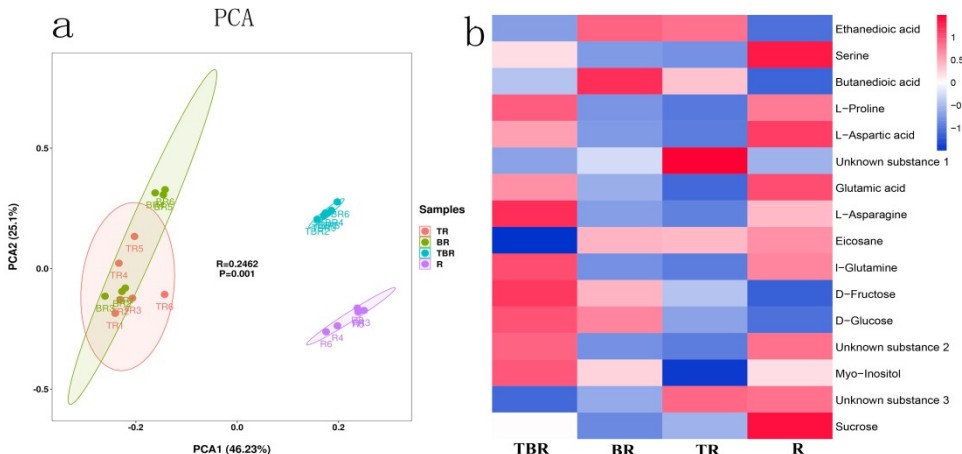

**Figure 3.** Graphical representation of changes in the metabolite profiles. (**a**) Principal component analyses (PCA) applied to tomato plants of different treatments according to their entire primary metabolome set of 16 metabolites; (**b**) heat-map of these 16 primary metabolites in different treatments. The treatments were as follows: R = *Ralstonia solanacearum* (RS-15) (as control); TR = *Trichoderma virens* (Tvien6) + RS-15; BR = *Bacillus velezensis* (X5) + RS-15; and TBR = Tvien6 + X5 + RS-15. The values of the heatmap are shown as the mean of six replications. Note: The color from red (value 1) to blue (value −1) represents the relative content of metabolites from high to low.

Based on the PCA, among the biocontrol agent treatment groups and the pathogen treatment group, TBR treatment was actually closer to R treatment than the TR or BR treatment group. As shown in Figure 3b, compared with the R treatment group, the TR and BR treatment groups each had a total of six metabolites with increased contents (such as D-fructose and D-glucose) and 10 metabolites with decreased contents (such as serine and L-aspartic acid), but the kinds and levels of contents of metabolites increased and decreased were not exactly the same in these two treatment groups; TBR treatment had a total of nine metabolites with increased contents (such as D-fructose and D-glucose), and the contents of seven metabolites were decreased (such as serine and glutamic acid).

There was a general trend toward higher D-fructose and D-glucose contents in the biocontrol agent treatment groups than in the pathogen only treatment group (Figure 4), and the order of the D-fructose (Figure 4a) and D-glucose (Figure 4b) contents between these treatment groups was TBR treatment > BR treatment > TR treatment > R treatment. Compared with the R treatment group (as a control), D-fructose was significantly increased in the TR (1.647-fold), BR (2.432-fold), and TBR (3.079-fold) treatment groups. Similarly, there was a significant accumulation of D-glucose in the TR (1.611-fold), BR (3.854-fold), and TBR (4.383-fold) treatment groups compared to the R treatment group. The results showed that applying Tvien6 or X5 individually can increase the D-fructose and D-glucose contents of tomato plants, and the combined application of Tvien6 and X5 can increase the accumulation of these metabolites.

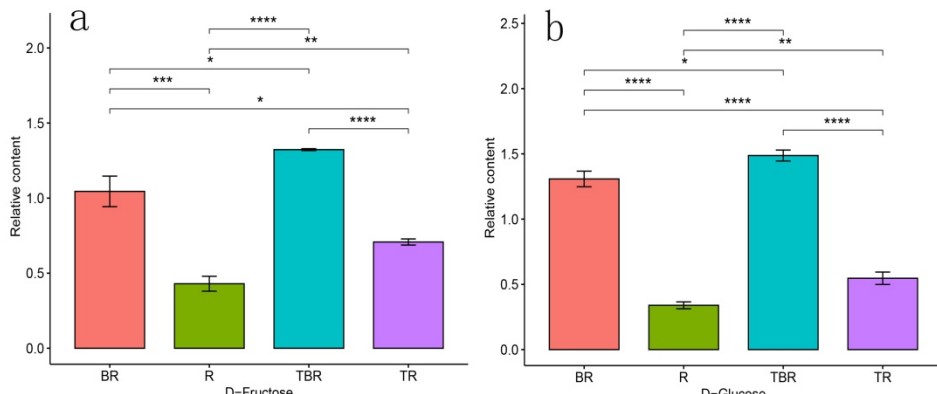

**Figure 4.** Effect of *Trichoderma virens* and *Bacillus velezensis* on the relative D-fructose and D-glucose content of tomato plants. The treatments were as follows: R = *Ralstonia solanacearum* (RS-15); TR = *Trichoderma virens* (Tvien6) + RS-15; BR = *Bacillus velezensis* (X5) + RS-15; and TBR = Tvien6 + X5 + RS-15. The relative D-fructose (**a**) and D-glucose (**b**) content are shown as the mean of six replications ± standard deviation (SD). * Represents significance; * represents that the *p* value is less than 0.05; ** represents that the *p* value is less than 0.01; *** represents that the *p* value is less than 0.001; and **** represents that the *p* value is less than 0.0001.

## 4. Discussion

*Trichoderma* spp. and *Bacillus* spp. are naturally abundant in soil and are best known as biocontrol agents for controlling soil-borne diseases. Currently, there are few studies on the combined application of *Trichoderma* and *Bacillus* to control tomato bacterial wilt disease. In this study, the combined application of Tvien6 with X5 showed increased antibacterial activity as a product of their synergistic action, resulting in reduced *Ralstonia solanacearum* growth and better control of disease compared with their individual application. Our current results also indicated that both Tvien6 and X5 could obviously promote tomato growth, however, their combination was more effective.

Increasing chlorophyll can improve the photosynthesis of plants, which can promote plant growth [46,63]. The biocontrol agents could increase the chlorophyll of plants, which is one of the reasons why biocontrol agents could promote plant growth [46,63,64]. Therefore, the reason for the TBR treatment group having the highest plant height and the heaviest plant weight of tomato may be that Tvien6 and X5 synergistically increase the chlorophyll content (SPAD value), which leads to the enhancement of tomato photosynthesis, which in turn more effectively promotes tomato growth.

In the present study, among the four treatment groups on days 0 to 9, the R treatment group had the lowest PPO, POD, and SOD activities, except that the PPO activity of the R treatment group was higher than that of the TR and BR treatment groups on day 3, and the SOD activity of the R treatment group was higher than that of the TR treatment group on day 9. In contrast, the TBR treatment group had the highest PPO, POD, and SOD activities, except that the POD activity of the TBR treatment group was lower than that of the BR treatment group on day 3. The current study demonstrated that the combination of Tvien6 and X5 was more effective in enhancing defensive enzyme (PPO, POD, and SOD) activities. Our results are analogous to Harish et al. (2009) [65], who showed that, compared with individual isolate applications, maximum PAL, POD, and PPO activities were induced in bananas challenged with the banana bunchy top virus that were treated with a mixture of the two isolates. Similarly, Mohan Kumar et al. (2015) [66] reported that tomato seeds treated with a combination of *Trichoderma harzianum* (OTPB3), *Bacillus subtilis* (OTPB1), and *Pseudomonas putida* (OPf1) had significantly reduced incidence of late blight and significantly increased PAL, PPO, and POD activities compared to treatment with OPf1, OTPB3, or OTPB1 alone. Similarly, maximum PAL, POD, and SOD activities in chickpea were induced in *Pseudomonas* (PHU094), *Trichoderma* (THU0816), and *Rhizobium* (RL091)

consortium-treated plants challenged with the pathogen compared to the single strain and dual strain consortia [57].

Increased PPO, POD, and SOD activities in plants have been directly linked to an increased defense response against phytopathogens [32,57,67,68]. PPO is a copper-containing enzyme that is important in plant defense, involved in catalyzing the oxidation of many phenolics to antimicrobial quinines and lignification of plant cells [35,57,69]. POD plays a key role in the second reaction in the process of reactive oxygen elimination by plants, and it can catalyze $H_2O_2$ to generate $H_2O$ and $\cdot OH$ with catalase (CAT) and Fe [70]. $O_2^-$ is toxic to organisms, and SOD is a first-line defensive enzyme contributing to resistance by producing $H_2O_2$ from $O_2^-$ [71]. SOD and POD could synergistically scavenge free radicals of plants [33,72]. In the present study, maximum SOD, PPO, and POD activities and the lowest disease incidences were generally observed in *Ralstonia solanacearum*-challenged tomatoes when Tvien6 and X5 were applied in combination; minimum SOD, PPO, and POD activities and the highest disease incidences were generally observed in the control (R treatment group). The possible reason for these results is that Tvien6 and X5 induce SOD and POD activities to scavenge free radicals in plant cells and induce PPO activities to promote the formation of antimicrobial quinines and lignin to protect against phytopathogen invasion, and defensive enzyme (PPO, POD, and SOD) activities were more effectively induced when the combination of Tvien6 and X5 was used.

Studying the changes in plant metabolites in response to exogenous microorganisms can help elucidate the mechanisms of interaction between plants and microorganisms. Fructose and glucose are important osmotic regulators and carbon and energy sources, and they play key roles in regulating gene expression, controlling plant growth and development, and stimulating plant responses to biotic and abiotic stresses [73–75]. Lecompte et al. (2017) [76] showed that fructose accumulation could reduce tomato susceptibility to *Botrytis cinerea* and recommended adjusting the fructose content to enhance plant defense capabilities. Similarly, Bogdanović et al. (2008) [77] indicated that fructose has a high capacity for scavenging superoxide, and increased fructose content represents an important antioxidative defense in plants under cold-related stress. Tomato glucose content changes under biotic stress; for example, the glucose content is upregulated in tomato stems and roots when parasitized by root-knot nematodes and *Egyptian broomrape*, respectively [48,78]. Glucose may also be related to plant disease resistance, and a study reported that *Arabidopsis* could limit the development of necrosis to reduce disease symptoms caused by *Botrytis cinerea* by increasing its glucose uptake rate [79].

The present study showed that the D-fructose and D-glucose contents were upregulated in tomatoes treated with Tvien6 or X5 and challenged with *Ralstonia solanacearum.* When the two biocontrol agents were applied in combination, D-fructose and D-glucose were significantly increased, indicating that Tvien6 and X5 synergistically enhanced the accumulation of D-fructose and D-glucose in tomatoes.

Overall, the reason for the lowest disease incidence in the TBR treatment group may be that Tvien6 and X5 synergistically induced the highest defensive enzyme activities and accumulated the highest D-fructose and D-glucose contents. The TBR treatment group had the highest plant height and the heaviest plant weight of tomato, which may depend on Tvien6 and X5 synergistically increasing the SPAD values.

## 5. Conclusions

PPO, POD, SOD, fructose, and glucose can improve the disease resistance of plants. In addition, chlorophyll content is positively correlated with plant growth. Furthermore, fructose and glucose are related to the growth of plants, and strong plants exhibit strong stress resistance. Our results showed that, compared with Tvien6 or X5 alone, the combination of Tvien6 and X5 was more effective in inhibiting *Ralstonia solanacearum* growth, promoting tomato plant growth, increasing leaf relative chlorophyll content (SPAD values), enhancing defensive enzyme (PPO, POD, and SOD) activities, and increasing D-fructose and D-glucose contents. These results suggest that the combined application of Tvien6 and

X5 could be considered as a promising approach for the control of tomato bacterial wilt disease due to the synergistic effects of the two biocontrol agents.

**Author Contributions:** Resources, methodology, funding acquisition, writing—review and editing, Y.Z.; Formal analysis, validation, L.Y.; Resources, data curation, writing—original draft, J.W.; Data curation, formal analysis, L.G.; and Conceptualization, supervision, project administration, funding acquisition, J.H. All authors have read and agreed to the published version of the manuscript.

**Funding:** This study was supported by Hainan Provincial Natural Science Foundation of China (318QN272); and Central Public-interest Scientific Institution Basal Research Fund for Chinese Academy of Tropical Agricultural Sciences (1630042019038; 2017hzs1J001).

**Institutional Review Board Statement:** Not applicable.

**Informed Consent Statement:** Not applicable.

**Data Availability Statement:** DNA sequences: GenBank numbers of the ITS, tef1, and rbp2 sequences of Tvien6 are MT102393, MT081440 and MT118254, respectively. GenBank number of the whole genome sequence of X5 is CP029473. All data used in this manuscript are available at this link: https://datadryad.org/stash/share/lwWmnQKytIwbSuZk9EMxinuvYj6ktie2RMqUoYDZLTk.

**Acknowledgments:** We are grateful to Chunxiao Li at Hainan University and AJE for helping to modify manuscript.

**Conflicts of Interest:** The authors declare no conflict of interest. The funders had no role in the design of the study; in the collection, analyses, or interpretation of data; in the writing of the manuscript, or in the decision to publish the results.

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
