# Peer review of "Synergistic Effect between Trichoderma virens and Bacillus velezensis on the Control of Tomato Bacterial Wilt Disease"

_horticulturae, doi:10.3390/horticulturae7110439_

Round 1

Reviewer 1 Report

This manuscript has potential to be highly cited due to the novel aspects mentioned in it. However there are some issues that require special attention before its consideration for its publication in Horticulturae.

These are particularly related to the statistics and the analysis of the samples. The standard deviations/ or standard error.

Figure 2 indicate how many plants and how many times were repeated the experiment.

The studies of SPAD require standard curves using chlorophyll as standard.

The metabolomics and the enzymatic activity assays is not indicated the number of samples used. 

Usually the figures and tables need to be placed after they are referred in the text and it needs to be fixed for the authors.

In figures 3 and 5 indicate the means of the bar in the lines or boxes. Are these bars standard or deviation errors? Also, is necessary to indicate the number of biological replicates. The small bars suggest that these data come from the measurement of replicates from a single biological replicate and not from different biological replicates. The authors need to clarify this and indicate the total number of biological replicates used.

For the enzymatic activity, the authors mentioned that each replicate was measured by triplicate (lines 138-139), but they did not mention how many biological replicates were used. If figure 3 represents the three measurements for a single replicate (biological sample?), the figure need to be change by including the other  biological replicates, but the authors needs to clarify if this is wrong.

Figure 4b. It is recommended to change the orientation of the treatment names. Also, indicate how many samples were analysed in the metabolomics and if these samples are represented in the heatmap.

Reviewer 2 Report

The manuscript ‘Synergistic effect between Trichoderma virens and Bacillus velezensis on the control of tomato bacterial wilt disease’ presents new data on the biological control against a serious bacterial disease on tomatoes. The two species are important for pathogens control in the soil. The synergistic effects of the combination of their using showed more efficient results than their separate applying.

The scientific information in the manuscript submitted for review is original. The new data expands the knowledge about the effect of Trichoderma virens and Bacillus velezensis on one of the important diseases on tomatoes causing by Ralstonia solanacearum. The results have great economic importance that will be used in plantations, especially with highly sensitive host plants.

Appropriate methods were used in the study. The attached illustrative plates reveal high-quality photos that are informative and useful.

The results of the study are presented correctly and are discussed. Knowledge of the effect of Trichoderma virens and Bacillus velezensis combine applying is of scientific interest to experts working in the field both of agronomy and phytopathology.

Author Response

Response to Reviewer 2 Comments

We sincerely thank you very much for reviewing our manuscript in detail and giving it a very high evaluation. Thank you again for your kindness.

Reviewer 3 Report

Introduction is very general and lack any specifics concerning the current state of knoledge. It is not enough to say that something was proved and then just put few references in the brackets. 

  1. l. 48-49: Maybe some more details here, which crop and which pathogen.
  2. l. 58-60: More suitable to M&M section, if left here more details should be added explaining why the method was crucial for this experiment and how it was applied by others. However, since it is not very new method, I would suggest removing or move to M&M.
  3. l. 65-66: as above.
  4. l. 62: Maybe some more detailed description here with examples, e.g. which group could be expected to be up/down-regulated. ´Changes in metabolites´ is very wide concept...

M&M 

  1. l. 77-79: Not really sure why those details are provided? From what I see gene sequences or genome sequence were not used anywhere in this study, e.g. in PCR or other molecular analysis. If some analysis were done to e.g. confirm the presence of the pathogen or identify that proper strain was used, it should be mentioned here.
  2. l. 85: should be ´per ml´ or ´ml^-1´ and not ´/ml`
  3. l. 126-127: write it as proper equation using special function in Word/Latex
  4. Enzyme assays: No proteins´ level measurements using Bradford to estimate enzymes activity in relation to proteins pool?
  5. l. 144 and further when centrifugation speed is specified: 12,000 xg not 12000 g
  6. l. 155: maybe ´in random order´ would suit better here, since I guess that what authors meant, since all the samples were processed in exact same way, or as I hope.
  7. l. 162: I don´t understand the meaning of the sentence, please reformulate.
  8. l. 168: You meant measurements/analysis or maybe derivatization?
  9. l. 174: ml min^-1
    please check the spelling of all units and transform it into the above scheme

Results: overall many claims in this section is not supported by the results and misleading for the reader. More detailed descriptions are needed, especially for enzyme activities and metabolites analysis.

  1. l. 188: The  antagonistic effect of Tvien6 is not really visible on this picture.
  2. l. 190-191: What is the point of showing this results for F. oxysporum in this manuscript? It is not discussed or used any further in the manuscript.
  3. l. 203-209: The statistical analysis seems to be wrong. The simple rule is that if two SEM error bars do overlap, and the sample sizes are equal or nearly equal, then you know that the P value is (much) greater than 0.05, so the difference is not statistically significant. Also if +/- 2 SEM overlap, sample means are not statistically different.
    Based on the values from the table: the only significant differences between treatments and control would be in DI and it is correctly stated. However, for all other parameters, R, BR and TR are not significantly different from each other, and TBR is most probably not different from BR. Therefore, the only significant difference is really between R and TBR.
  4. Figure 2 (lower half): The picture of plants in pots may be misleading suggesting that TBR treated plants are almost twice as big as R plants, which is very much not what appeares from the results presented in the Table 1.
  5. l. 228-229: PPO values for BR decrease only after 6 days, however rather not significantly.
  6. l. 231-232: For BR and TR, POD values increased after 3 days, but then decreased after 6 days. While for R and TBR there is constant increase in comparison with 0D. Therefore putting it all together in one sentence like that suggest the same trend in all groups.
  7. l. 236-237: not according to the Figure 3 below...
  8. Fig. 3: No explanation what error bars means: SE or SD?
  9. l. 250: why those 16 metabolites were chosen in a first place?
  10. l. 255-256: I wouldn´t say that clearly different profiles were observed for 4 treatments. Many similarities are observed between TR and BR, and interestingly between R and TBR, based both on heatmap and PCA analysis where TBR samples are actually grouped closer to R samples than overlaping TR and BR. It should be described here.
  11. l. 260-261: Fig. 4b, What this heatmap actually represents? It shows positive and negative values suggesting ratio, but towards what? Not towards control (R) since it´s also there?
  12. Fig. 5: What ´Relative content´ means? Relative towards what? 

Discussion - more thorough discussion of obtained results is needed. Authors should at least try to link all obtained results to explain why plants treated with TBR where less affected by the pathogen. Also discussion with others results is almost non-existing.

  1. l. 289: no positive effect of Tvien6 on parameters other than DI was showed in the manuscript.
  2. l. 301-303: Possible mechanism explaining why high activity of those enzymes resulted in lower DI of TBR plants should be explained here. Also in terms when each enzyme showed peak activity, etc.

Conclusions

  1. l. 340: it actually just shows what is the synergistic effect of these two agents on plant, and not explanation why. To answer that rather analysis of microbes should be performed, like transcriptome or metabolome analysis. 

Reviewer 4 Report

Review Comments for Manuscript ID horticulturae-1307977 “Synergistic effect between Trichoderma virens and Bacillus velezensis on the control of tomato bacterial wilt disease”

Line 78: Replace “rbp2” with “rpb2”.

Line 79-81: “the pathogen…. symptoms”. How the authors confirm that the pathogen isolated from naturally infecting tomatoes was Ralstonia and no other bacterial pathogen. Did you confirm it through sequencing? Please clarify.

Line 92-103: Did statistical analysis was performed to confirm the effect of Trichoderma virens and Bacillus velezensis culture filtrates on Ralstonia. Please include the statistical analysis results. Please specify control.

Line 114: Specify BR, TR and TBR. Specify control.

Line 125: Remove parenthesis

Line 123: Specify if the statistical analysis was performed. Specify control

Line 137-150: Write statistical analysis was performed for these experiments, Include the statement here. Specify control

Line 203-310: Include the results from the statistical analysis in the text. Indicate in the text if there was any significant difference in the treatments. Specify control and results compared to control

Line 222-243: Indicate in the text if there is statistical differences in the enzymatic assays in different treatments. Specify control and significant difference compared to control

Line 249-274: Include the statement for the statistical analysis and whether the results were statistically significant from one another and compared to control.

General- Specify control in the manuscript clearly.

Round 2

Reviewer 3 Report

Introduction:

  • l. 76-82: No research so far concerning metabolism changes for plants infected with Ralstonia solanacearum or at least other bacterial pathogen in tomato or other related plants?
    This paragraph is really weak, seems like just some random information was added, not really related with the research subject.

M&M:

  • l. 92-94: Still no explanation for the reader what is the point of adding such information for Tvien6.
  • l. 97: Were those primers designed by the authors? If not there need to be references for each pair?
  • l. 190-191: This is very strangely formulated. Why don't just write: `Methanol was used as a blank sample´? Or did I understand it wrongly?

Results:

  • l. 243: TR did not significantly increase SPAD in comparison with R (the control)
  • Enzyme assays: This is very messy description of results and hard to follow. Authors jump in-between description of different treatments in pretty random way. It is hard to understand why only some results are mentioned and other omitted, e.g. decrease of POD activity at day 9 among others.
    • l. 259: Shouldn´t it be ´BR decrease only after 9 days´?
    • l. 259-260: was it significant or not according to the statistical analysis?
    • l. 271: Not really a peak at day 3 for TR as there is hardly any difference comparing to day 0.
    • Fig. 2: It is hard to read the figure. Using different colors for each column would make it easier to follow.
  • l. 294-295: The whole point is that the analysis did not show clearly that 4 treatments resulted in different metabolite profiles, as is described in the next sentence.
  • l. 296-299: Have authors seriously copied my exact comment on this paragraph and make it a part of their manuscript? I mean, come on... 
  • Fig. 3: There is still no explanation what the heat-map is showing? E.g. what it means that Myo-inositol value for TR is lower than -1? There is no explanation under the figure towards what? It should be explained what the legend bar showing values range and colors means. At the moment, reader might be confused what is shown there.
  • Fig. 4: it should be explained in the description relative towards what? 

Discussion: no significant changes were done in comparison with the previous version, apart from adding one or two sentences and mixing the structure a bit. 
